# OmniSCV: An Omnidirectional Synthetic Image Generator for Computer Vision

**DOI:** 10.3390/s20072066

**Published:** 2020-04-07

**Authors:** Bruno Berenguel-Baeta, Jesus Bermudez-Cameo, Jose J. Guerrero

**Affiliations:** Instituto de Investigación en Ingeniería de Aragón, Universidad de Zaragoza, 50018 Zaragoza, Spain; bermudez@unizar.es (J.B.-C.); jguerrer@unizar.es (J.J.G.)

**Keywords:** computer vision, omnidirectional cameras, virtual environment, deep learning, non-central systems, image generator, semantic label

## Abstract

Omnidirectional and 360° images are becoming widespread in industry and in consumer society, causing omnidirectional computer vision to gain attention. Their wide field of view allows the gathering of a great amount of information about the environment from only an image. However, the distortion of these images requires the development of specific algorithms for their treatment and interpretation. Moreover, a high number of images is essential for the correct training of computer vision algorithms based on learning. In this paper, we present a tool for generating datasets of omnidirectional images with semantic and depth information. These images are synthesized from a set of captures that are acquired in a realistic virtual environment for Unreal Engine 4 through an interface plugin. We gather a variety of well-known projection models such as equirectangular and cylindrical panoramas, different fish-eye lenses, catadioptric systems, and empiric models. Furthermore, we include in our tool photorealistic non-central-projection systems as non-central panoramas and non-central catadioptric systems. As far as we know, this is the first reported tool for generating photorealistic non-central images in the literature. Moreover, since the omnidirectional images are made virtually, we provide pixel-wise information about semantics and depth as well as perfect knowledge of the calibration parameters of the cameras. This allows the creation of ground-truth information with pixel precision for training learning algorithms and testing 3D vision approaches. To validate the proposed tool, different computer vision algorithms are tested as line extractions from dioptric and catadioptric central images, 3D Layout recovery and SLAM using equirectangular panoramas, and 3D reconstruction from non-central panoramas.

## 1. Introduction

The great amount of information that can be obtained from omnidirectional and 360° images makes them very useful. Being able to obtain information from an environment using only one shot makes these kinds of images a good asset for computer vision algorithms. However, due to the distortions they present, it is necessary to adapt or create special algorithms to work with them. New computer vision and deep-learning-based algorithms have appeared to take advantage of the unique properties of omnidirectional images. Nevertheless, for a proper training of deep-learning algorithms, big datasets are needed. Existing datasets are quite limited in size due to the manual acquisition, labeling and post-processing of the images. To make faster and bigger datasets, previous works such as [1,2,3,4,5] use special equipment to obtain images, camera pose, and depth maps simultaneously from indoor scenes. These kinds of datasets are built from real environments, but need post-processing of the images to obtain semantic information or depth information. Tools like LabelMe [6] and new neural networks such as SegNet [7] can be used to obtain automatic semantic segmentation from the real images obtained in the previously mentioned datasets, yet without pixel precision. Datasets such as [8,9] use video sequences from outdoor scenes to obtain depth information for autonomous driving algorithms. In addition, for these outdoor datasets, neural networks are used to obtain semantic information from video sequences [10,11], in order to speed up and enlarge the few datasets available.

Due to the fast development of graphic engines such as Unreal Engine [12], virtual environments with realistic quality have appeared. To take advantage of this interesting property, simulators such as CARLA [13] and SYNTHIA [14] recreate outdoor scenarios in different weather conditions to create synthetic datasets with labeled information. If we can define all the objects in the virtual environment, it is easier to create a semantic segmentation and object labeling, setting the camera pose through time and computing the depth for each pixel. These virtual realistic environments have helped to create large datasets of images and videos, mainly from outdoor scenarios, dedicated to autonomous driving. Other approaches use photorealistic video games to generate the datasets. Since these games already have realistic environments designed by professionals, many different scenarios are recreated, with pseudo-realistic behaviors of vehicles and people in the scene. Works such as [15] use the video game Grand Theft Auto V (GTA V) to obtain images from different weather conditions with total knowledge of the camera pose, while [16,17] also obtaining semantic information and object detection for tracking applications. In the same vein, [18,19] obtain video sequences with semantic and depth information for the generation of autonomous driving datasets in different weather conditions and through different scenarios, from rural roads to city streets. New approaches such as the OmniScape dataset [20] uses virtual environments such as CARLA or GTA V to obtain omnidirectional images with semantic and depth information in order to create datasets for autonomous driving.

However, most of the existing datasets have only outdoors images. There are very few synthetic indoor datasets [21] and most of them only have perspective images or equirectangular panoramas. Fast development of computer vision algorithms demands ever more omnidirectional images and that is the gap between the resources that we want to fill in this work. In this work we present a tool to generate image datasets from a huge diversity of omnidirectional projection models.

We focus not only on panoramas, but also on other central projections, such as fish-eye lenses [22,23], catadioptric systems [24] and empiric models such as Scaramuzza’s [25] or Kannala–Brandt’s [26]. Our novelty resides in the implementation of different non-central-projection models, such as non-central panoramas [27] or spherical [28] and conical [29] catadioptric systems in the same tool.

The composition of the images is made in a virtual environment from Unreal Engine, making camera calibration and image labeling easier. Moreover, we implement several tools to obtain ground-truth information for deep-learning applications, for example layout recovery or object detection.

The main contributions of this work can be summarized as follows:Integrating in a single framework several central-projection models from different omnidirectional cameras as panoramas, fish-eyes, catadioptric systems, and empiric models.Creating the first photorealistic non-central-projection image generator, including non-central panoramas and non-central catadioptric systems.Devise a tool to create datasets with automatic labeled images from photorealistic virtual environments.Develop automatic ground-truth generation for 3D layout recovery algorithms and object detection.

The next section of this work is divided in 4 main parts. In the first one, Section 2.1, we introduce the virtual environment in which we have worked. Section 2.2 presents the mathematical background of the projection models implemented and in Section 2.3 and Section 2.4 we explain how the models are implemented.

## 2. Materials and Methods

The objective of this work is to develop a tool to create omnidirectional images enlarging existing datasets or making new ones to be exploited by computer vision algorithms under development. For this purpose, we use virtual environments, such as Unreal Engine 4 [12], from where we can get perspective images to compose 360° and omnidirectional projections. In these environments, we can define the camera (pose, orientation and calibration), the layout, and the objects arranged in the scene, making it easier to obtain ground-truth information.

The proposed tool includes the acquisition of images from a virtual environment created with Unreal Engine 4 and the composition of omnidirectional and 360 images from a set of central and non-central camera systems. Moreover, we can acquire photorealistic images, semantic segmentation on the objects of the scene or depth information from each camera proposed. Furthermore, given that we can select the pose and orientation of the camera, we have enough information for 3D-reconstruction methods.

### 2.1. Virtual Environments

Virtual environments present a new asset in the computer vision field. These environments allow the generation of customized scenes for specific purposes. Moreover, great development of computer graphics has increased the quality and quantity of graphic software, obtaining even realistic renderings. A complex modeling of the light transport and its interaction with objects is essential to obtain realistic images. Virtual environments such as POV-Ray [30] and Unity [31] allow the generation of customized virtual environments and obtain images from them. However, they do not have the realism or flexibility in the acquisition of images we are looking for. A comparative of images obtained from acquisitions in POV-Ray and acquisitions in Unreal Engine 4 is presented in Figure 1.

The virtual environment we have chosen is Unreal Engine 4, (UE4) [12], which is a graphic engine developed by EpicGames (https://www.epicgames.com). Being an open-source code has allowed the development of a great variety of plugins for specific purposes. Furthermore, realistic graphics in real time allows the creation of simulations and generation of synthetic image datasets that can be used in computer vision algorithms. Working on virtual environments makes easier and faster the data acquisition than working on the field.

In this work, we use UE4 with UnrealCV [32], which is a plugin designed for computer vision purposes. This plugin allows client–server communication with UE4 from external Python scripts (see Figure 2) which is used to automatically obtain many images. The set of available functions includes commands for defining and operating virtual cameras; i.e., fixing the position and orientation of the cameras and acquiring images. As can be seen in Figure 3, the acquisition can obtain different kinds of information from the environment (RGB, semantic, depth or normals).

However, the combination of UE4+UnrealCV only allows perspective images, so it is necessary to find a way to obtain enough information about the environment to obtain omnidirectional images and in particular to build non-central images. For central omnidirectional images, the classical adopted solution is the creation of a cube map [33]. This proposal consists of taking 6 perspective images from one position so we can capture the whole environment around that point. We show that this solution only works for central projections, where we have a single optical center that matches with the point where the cube map has been taken. Due to the characteristics of non-central-projection systems, we make acquisitions in different locations, which depend on the projection model, to compose the final image.

### 2.2. Projection Models

In this section, we introduce the projection models for the different cameras that are implemented in the proposed tool. We are going to explain the relationship between image–plane coordinates and the coordinates of the projection ray in the camera reference. We distinguish two types of camera models: central-projection camera models and non-central-projection camera models. Among the **central** projection cameras, we consider:Panoramic images: *Equirectangular* and *Cylindrical*Fish-eye cameras, where we distinguish diverse lenses: *Equi-angular, Stereographic, Equi-solid angle, Orthogonal*Catadioptric systems, where we distinguish different mirrors: *Parabolic* and *Hyperbolic*Scaramuzza model for revolution symmetry systemsKannala–Brandt model for fish-eye lenses

Among the **non-central** projection cameras, we consider:Non-central panoramasCatadioptric systems, where we distinguish different mirrors: *Spherical* and *Conical*

#### 2.2.1. Central-Projection Cameras

Central-projection cameras are characterized by having a unique optical center. That means that every ray coming from the environment goes through the optical center to the image. Among omnidirectional systems, panoramas are the most used in computer vision. Equirectangular panoramas are 360°-field-of-view images that show the whole environment around the camera. This kind of image is useful to obtain a complete projection of the environment from only one shot. However, this representation presents heavy distortions in the upper and lower part of the image. That is because the equirectangular panorama is based on spherical coordinates. If we take the center of the sphere as the optical center, we can define the ray that comes from the environment in spherical coordinates (θ,ϕ). Moreover, since the image plane is an unfolded sphere, each pixel can be represented in the same spherical coordinates, giving a direct relationship between the image plane and the ray that comes from the environment. This relationship is described by:(1)$$\left(\theta ,\varphi \right)=\left(\left(2x/x_{max}-1\right)\pi ,\left(1/2-y/y_{max}\right)\pi \right),$$
where (x,y) are pixel coordinates and $\left(x_{max},y_{max}\right)$ the maximum value, i.e., the image resolution.

In the case of cylindrical panoramas, the environment is projected into the lateral surface of a cylinder. This panorama does not have a 360° practical field of view, since the perpendicular projection to the lateral surface of the environment cannot be projected. However, we can achieve up to 360° on the horizontal field of view, $FOV_{h}$, and theoretical 180° on the vertical field of view, $FOV_{v}$, that usually is reduced from 90° to 150° for real applications. We can describe the relationship between the ray that comes from the environment and the image plane as:(2)$$\left(\theta ,\varphi \right)=\left(\left(2x/x_{max}-1\right)FOV_{h}/2,\left(1-2y/y_{max}\right)FOV_{v}/2\right)$$

Next, we introduce the fish-eye cameras [22]. The main characteristic of this kind of camera is the wide field of view. The projection model for this camera system has been obtained in [34], where a unified model for revolution symmetry cameras is defined. This method consists of the projection of the environment rays into a unit radius sphere. The intersection between the sphere and the ray is projected into the image plane through a non-linear function $\hat{r}=h\left(\phi \right)$, which depends on the angle of the ray and the modeled fish-eye lens. In Table 1
h(ϕ) for the lenses implemented in this work is defined.

For catadioptric systems, we use the sphere model, presented in [24]. As in fish-eye cameras, we have the intersection of the environment’s ray with the unit radius sphere. Then, through a non-linear function, we project the intersected point into a normalized plane. The non-linear function, h(x), depends on the mirror we are modeling. The final step of this model projects the point in the normalized plane into the image plane with the calibration matrix H${}_{c}$, defined as H${}_{c}$ = K${}_{c}$R${}_{c}$M${}_{c}$, where K${}_{c}$ is the calibration matrix of the perspective camera, R${}_{c}$ is the rotation matrix of the catadioptric system and M${}_{c}$ defines the behavior of the mirror (see Equation (Equation 3)).
(3)$$K_{c}=\left(\begin{array}{ccc} f_{x} & 0 & u_{0}\\ 0 & f_{y} & v_{0}\\ 0 & 0 & 1 \end{array}\right);M_{c}=\left(\begin{array}{ccc} \Psi -\xi & 0 & 0\\ 0 & \xi -\Psi & 0\\ 0 & 0 & 1 \end{array}\right),$$
where $\left(f_{x},f_{y}\right)$ are the focal length of the camera, and $\left(u_{0},v_{0}\right)$ the coordinates of the optical center in the image, the parameters Ψ and ξ represent the geometry of the mirror and are defined in Table 2, *d* is the distance between the camera and the mirror and *2p* is the semi-latus rectum of the mirror.

The last central-projection models presented in this work are the Scaramuzza and Kannala–Brandt models. Summarizing [25,26], these empiric models represent the projection of a 3D point into the image plane through non-lineal functions.

In the Scaramuzza model, the projection is represented by λp=λg(u″)=PX. The non-lineal function, g(u″)=(u″,v″,f(u″,v″)), is defined by the image coordinates and a n-grade polynomial function $f\left(u\prime \prime ,v\prime \prime \right)=a_{0}+a_{1}\rho +a_{2}\rho^{2}+\cdots +a_{N}\rho^{N}$, where ρ is defined as the distance of the pixel to the optical center in the image plane, and $\left[a_{0},a_{1},\cdots ,a_{N}\right]$ are calibration parameters of the modeled camera.

In the Kannala–Brandt camera model, the forward projection model is represented as:(4)$$\pi \left(x,i\right)=\left[\begin{array}{c} f_{x}d\left(\theta \right)\frac{x}{r})\\ f_{y}d\left(\theta \right)\frac{y}{r}) \end{array}\right]+\left[\begin{array}{c} c_{x}\\ c_{y} \end{array}\right],$$
where $r=\sqrt{x^{2}+y^{2}}$, $\left(c_{x},c_{y}\right)$ are the coordinates of the optical center and d(θ) is the non-linear function which is defined as $d\left(\theta \right)=\theta +k_{1}\theta^{3}+k_{2}\theta^{5}+k_{3}\theta^{7}+k_{4}\theta^{9}$, where θ=arctan(r/z) and $\left[k_{1},k_{2},k_{3},k_{4}\right]$ are the parameters that characterize the modeled camera.

#### 2.2.2. Non-Central-Projection Cameras

Central-projection cameras are characterized by the unique optical center. By contrast, in non-central-projection models, we do not have a single optical center for each image. For the definition of non-central-projection models, we use Plücker coordinates. In this work, we summarize the models; however, a full explanation of the models and Plücker coordinates is described in [35].

Non-central panoramas have similarities with equirectangular panoramas. The main difference with central panoramas is that each column of the non-central panorama shares a different optical center. Moreover, since central panoramas are the projection of the environment into a sphere, non-central panoramas are the projection into a semi-toroid, as can be seen in Figure 4a. The optical center of the image is distributed in the trajectory of centers of the semi-toroid. This trajectory is defined as a circle, dashed line in Figure 4a, whose center is the revolution axis. The definition of the rays that go to the environment from the non-central system is obtained in [27], given as the result of Equation (Equation 5). The parameter $R_{c}$ is the radius of the circle of optical centers and (θ,φ) are spherical coordinates in the coordinate system.
(5)$$\Xi ={\left(\begin{array}{cc} \xi & \bar{\xi } \end{array}\right)}^{T}={\left(\begin{array}{c} \sin\varphi \cos\theta ,\sin\varphi \sin\theta ,\cos\varphi ,R_{c}\sin\varphi \sin\theta ,-R_{c}\sin\varphi \cos\theta ,0 \end{array}\right)}^{T}$$

Finally, spherical and conical catadioptric systems are also described by non-central-projection models. Just as with non-central panoramas, a full explanation of the model can be found on [28,29].

Even though the basis of non-central and central catadioptric systems is the same, we take a picture of a mirror from a perspective camera, the mathematical model is quite different. As in non-central panoramas, for spherical and conical mirror we also use the Plücker coordinates to define projection rays; see Figure 4b. For conical catadioptric systems, we define $Z_{r}=Z_{c}+R_{c}\cot\phi$, where $Z_{c}$ and $R_{c}$ are geometrical parameters and cotϕ=(z+rtan2τ)/(ztan2τ−r), where τ is the aperture angle of the cone. When these parameters are known, the 3D ray in Plücker coordinates is defined by:(6)$$\Xi ={\left(\begin{array}{cc} \xi & \bar{\xi } \end{array}\right)}^{T}={\left(\begin{array}{c} \sin\varphi \cos\theta ,\sin\varphi \sin\theta ,\cos\varphi ,-Z_{r}\sin\varphi \sin\theta ,Z_{r}\sin\varphi \cos\theta ,0 \end{array}\right)}^{T}$$

For the spherical catadioptric system, we define the geometric parameters as $Z_{s}=Z_{m}+R_{s}$, $Z_{rel}=Z_{s}/R_{s}$, $r^{2}=x^{2}+y^{2}$ and $\rho^{2}=x^{2}+y^{2}+z^{2}$. Given the coordinates at a point of the image plane, the Equation (Equation 7) defines the ray that reflects on the mirror:(7)$$\Xi ={\left(\begin{array}{cc} \xi & \bar{\xi } \end{array}\right)}^{T}={\left(\begin{array}{c} -x\delta ,y\delta ,-\zeta ,\epsilon yZ_{s},\epsilon xZ_{s},0 \end{array}\right)}^{T},$$
where $\delta =2r^{2}Z_{rel}^{4}-2z\sqrt{\gamma }Z_{rel}^{2}-3\rho^{2}Z_{rel}^{2}+\rho^{2}$; $\epsilon =\left(-r^{2}+z^{2}\right)Z_{rel}^{2}+2\sqrt{\gamma }z+\rho^{2}$; $\zeta =2r^{2}zZ_{rel}^{4}-z\rho^{2}Z_{rel}^{2}-2\sqrt{\gamma }\left(-r^{2}Z_{rel}^{2}+\rho^{2}\right)-z\rho^{2}$ and $\gamma =\left(-r^{2}Z_{rel}^{2}+\rho^{2}\right)Z_{rel}^{2}$.

### 2.3. Central Cameras Simulator

In this section, we describe the simulator, the interaction with UnrealCV and how are the projection models are implemented. The method to obtain omnidirectional images can be summarized in two steps:**Image acquisition**: the first step is the interaction with UE4 through UnrealCV to obtain the cube map from the virtual environment.**Image composition**: the second step is creating the final image. In this step we apply the projection models to select the information of the environment that has been acquainted in the first step.

For central-projection images, the two steps are independent from each other. Once we have a cube map, we can build any central-projection image from that cube map. However, for non-central-projection images, the two steps are mixed. We need to compute where the optical center is for each pixel and make the acquisition for that pixel. Examples of the images obtained for each projection model can be found in the Appendix A.

#### 2.3.1. Image Acquisition

The image acquisition is the first step to build omnidirectional images. In this step we must interact with UE4 through UnrealCV using Python scripts. Camera pose and orientation, acquisition field of view and mode of acquisition are the main parameters that we must define in the scripts to give the commands to UnrealCV.

In this work, we call cube map to the set of six perspective images that models the full 360° projection of the environment around a point; concept introduced in [33]. Notice that 360° information around a point can be projected into a sphere centered in this point. Composing a sphere from perspective images requires a lot of time and memory. Simplifying the sphere into a cube, as seen in Figure 5a, we have a good approximation of the environment without losing information; see Figure 5b. We can make this affirmation since the defined cube is a smooth atlas of the spherical manifold $S^{2}$ embedded in $R^{3}$.

To create central-projection systems, the acquisition of each cube map must be done from a single location. Each cube map is the representation of the environment from one position—the optical center of the omnidirectional camera. That is why we use UE4 with UnrealCV, where we can define the camera pose easily. Moreover, the real-time renderings of the realistic environments allows fast acquisitions of the perspective images to build the cube maps. Nevertheless, other virtual environments can be used to create central-projection systems whenever the cube map can be built with these specifications.

Going back to UnrealCV, the plugin gives us different kinds of capture modes. For our simulator, we have taken 3 of these modes: **lit, object mask** and **depth**.

In the **lit** mode, UnrealCV gives a photorealistic RGB image of the virtual environment. The degree of realism must be created by the designer of the scenario. The second is the **object mask** mode. This mode gives us semantic information of the environment. The images obtained have a colored code to identify the different objects into the scene. The main advantage of this mode is the pixel precision for the semantic information, avoiding the human error in manual labeling. Moreover, from this capture mode, we can obtain ground-truth information of the scene and create specific functions to obtain ground-truth data for computer vision algorithms, as layout recovery or object detection. The third mode is **depth**. This mode gives a data file where we have depth information for each pixel of the image. For the implementation of this mode in the simulator, we keep the exact data information and compose a depth image in grayscale.

#### 2.3.2. Image Composition

Images from central-projection cameras are composed from a cube map acquired in the scene. The composition of each image depends on the projection model of the camera, but they follow the same pattern.

The Algorithm 1 shows the steps in the composition of central-projection images. Initially, we get the pixel coordinates from the destination image that we want to build. Then, we compute the spherical coordinates for each pixel through the projection model of the camera we are modeling. With the spherical coordinates we build the vector that goes from the optical center of the camera to the environment. Then, we rotate this vector to match the orientation of the camera into the environment. The intersection between this rotated vector and the cube map gives the information of the pixel (color, label, or distance). In contrast to the coordinate system of UE4, which presents a left-handed coordinate system and different rotations for each axis (see Figure 6a), our simulator uses a right-handed coordinate system and spherical coordinates, as shown in Figure 6b. Changing between the coordinate systems is computed internally in our tool to keep mathematical consistency between the projection models and the virtual environment.
**Algorithm 1:** Central-projection composition   **Input**: Set-up of final image (Camera, Resolution, type, ...)   Load cube map   **while**
*Go through final image*
**do**       get pixel coordinates       compute spherical coordinates -> Apply projection model of omnidirectional camera       compute vector/light ray -> Apply orientation of the camera in the environment       get pixel information -> Intersection between light ray and cube map   **end**

**Equirectangular** panoramas have an easy implementation in this simulator since equirectangular images are the projection of the environment into a sphere. Then, the destination image can be defined in spherical coordinates directly from the pixel coordinates as:(8)$$\theta =\left(\frac{2u}{u_{max}}-1\right)\pi ;\;\varphi =\left(1/2-\frac{v}{v_{max}}\right)\pi ,$$
where these equations define the range of the spherical coordinates as −π<θ<π and −π/2<φ<π/2.

In the **cylindrical** model we also use spherical coordinates, but defined with the restrictions of the cylindrical projection. In the definition of the Equation (Equation 9) we can see that two new parameters appear. The $FOV_{h}$ parameter represents the horizontal field of view of the destination image, which can go up to 360°, and can be changed by the simulator user. The $FOV_{v}$ parameter models the height of the lateral surface of the cylinder from the optical center point of view.
(9)$$\theta =\left(\frac{2u}{u_{max}}-1\right)\frac{FOV_{h}}{2};\;\varphi =\left(1-\frac{2v}{v_{max}}\right)\frac{FOV_{v}}{2},$$
where the range of the spherical coordinates are $-FOV_{h}/2<\theta <FOV_{h}/2$ and $-FOV_{v}/2<\varphi <FOV_{v}/2$.

In **fish-eye** cameras we have a revolution symmetry, so we use polar coordinates. We transform the pixel coordinates into polar coordinates with the next equation:(10)$$\left(\hat{r},\theta \right)=\left(\sqrt{{\left(u-u_{0}\right)}^{2}+{\left(v-v_{0}\right)}^{2}},\arctan\left(\left(u_{0}-u\right)/\left(v-v_{0}\right)\right)\right),$$
where we define $\left(u_{0},v_{0}\right)=\left(u_{max}/2,v_{max}/2\right)$.

Given this definition, the range of the polar coordinates is $0<\hat{r}<\sqrt{(u_{max}^{2}+v_{max}^{2})/4}$ and −π<θ<π. However, we crop ${\hat{r}}_{max}=min\left(u_{max}/2,v_{max}/2\right)$ in order to constrain the rendered pixels in the camera field of view. After obtaining the polar coordinates, we get the spherical coordinates for each pixel through the projection model. In Table 3 we have the relationship between the polar coordinate $\hat{r}$ and the spherical coordinate ϕ given by the projection model of fish-eye cameras.

where *f* is the focal length of the fish-eye camera.

On **catadioptric systems**, we define the parameters ξ and η that model the mirror as shown in Table 4. We use polar coordinates to select what pixels of the image are rendered, but we apply the projection model on the pixel coordinates directly.

For computing the set of rays corresponding to the image pixels, we use the back-projection model described in [36]. The first step is projecting the pixel coordinates into a 2D projective space: $p={\left(u,v\right)}^{T}\overset{}{\to }\hat{v}={\left(\hat{x},\hat{y},\hat{z}\right)}^{T}$. Then, we re-project the point into the normalized plane with the inverse calibration matrix of the catadioptric system as $\bar{v}=H_{c}^{-1}\hat{v}$, (see Section 2.2). Finally, through the inverse of the non-linear function h(x), shown in the Equation (Equation 11), we can obtain the coordinates of the ray that goes from the catadioptric system to the environment.
(11)$$v=h^{-1}\left(\bar{v}\right)=\left(\begin{array}{c} x\\ y\\ z \end{array}\right)=\left(\begin{array}{c} \bar{x}\frac{\bar{z}\xi +\sqrt{{\bar{z}}^{2}+\left(1-\xi^{2}\right)\left({\bar{x}}^{2}+{\bar{y}}^{2}\right)}}{{\bar{x}}^{2}+{\bar{y}}^{2}+{\bar{z}}^{2}}\\ \bar{y}\frac{\bar{z}\xi +\sqrt{{\bar{z}}^{2}+\left(1-\xi^{2}\right)\left({\bar{x}}^{2}+{\bar{y}}^{2}\right)}}{{\bar{x}}^{2}+{\bar{y}}^{2}+{\bar{z}}^{2}}\\ \bar{z}\frac{\bar{z}\xi +\sqrt{{\bar{z}}^{2}+\left(1-\xi^{2}\right)\left({\bar{x}}^{2}+{\bar{y}}^{2}\right)}}{{\bar{x}}^{2}+{\bar{y}}^{2}+{\bar{z}}^{2}}-\xi \end{array}\right)$$

These projection models give us an oriented ray that comes out of the camera system. This ray is expressed in the camera reference. Since the user of the simulator can change the orientation of the camera in the environment in any direction, we need to change the reference system of the ray to the world reference. First, we rotate the ray from the camera reference to the world reference. Then we rotate again the ray in the direction of the camera inside the world reference. After these two rotations, we have changed the reference system of the ray from the camera to the world reference, taking into account the orientation of the camera in the environment. Once the rays are defined, we get the information from the environment computing the intersection of each ray with the cube map. The point of intersection has the pixel information of the corresponding ray.

### 2.4. Non-central Camera Simulator

The simulator for non-central cameras is quite different from the central camera one. In this case, we can neither use only a cube map to build the final images nor save all the acquisitions needed. The main structure proposed to obtain non-central-projection images is shown in Algorithm 2. Since we have different optical centers for each pixel in the final image, we group the pixels sharing the same optical center, reducing the number of acquisitions needed to build it. The non-central systems considered group the pixel in different ways, so the implementations are different.
**Algorithm 2:** Non-central-projection composition   **Input**: Set-up of final image (Camera, Resolution, type, ...)   **while**
*Go through final image*
**do**       get pixel coordinates       compute optical center       make image acquisition -> Captures from UnrealCV       compute spherical coordinates -> Apply projection model       get pixel information -> Intersection with acquired images   **end**

From the projection model of non-central panoramas, we get that pixels with the same *u* coordinate share the same optical center. For each coordinate *u* of the image, the position for the image acquisition is computed. For a given center ${\left(X_{c},Y_{c},Z_{c}\right)}^{T}$, and radius $R_{c}$, of the non-central system, we have to compute the optical center of each *u* coordinate.
(12)$${\left(x_{oc},y_{oc},z_{oc}\right)}^{T}={\left(X_{c},Y_{c},Z_{c}\right)}^{T}+R_{c}{\left(\cos\theta \cos\phi ,\sin\theta ,\cos\theta \sin\phi \right)}^{T}$$
(13)$$\theta =\left(\frac{2u}{u_{max}}-1\right)\pi$$

To obtain each optical center, we use Equation (Equation 12), where θ is computed according to Equation (Equation 13) and ϕ is the pitch angle of the non-central system. Once we have obtained the optical center, we make the acquisition in that location, obtaining a cube map. Notice that this cube map only allows the obtaining of information for the location it has acquired. This means that for every optical center of the non-central system, we must make a new acquisition.
(14)$$\varphi =\left(1/2-\frac{v}{v_{max}}\right)\pi$$

Once the acquisition is obtained from the correct optical center, we compute the spherical coordinates to cast the ray into the acquired images. From the Equation (Equation 13) we already have one of the coordinates; the second is obtained from the Equation (Equation 14).

In non-central catadioptric systems, pixels sharing the same optical center are grouped in concentric circles. That means we go through the final image from the center to the edge, increasing the radius pixel by pixel. For each radius, we compute the parameters $Z_{r}$ and cotϕ as in Table 5, which depend on the mirror we want to model (see Section 2.2). The parameter $Z_{r}$ allows us to compute the optical center from where we acquire a cube map of the environment.
(15)$$\left(u,v\right)=\left(u_{max}/2+r\cos\theta ,v_{max}/2-r\sin\theta \right)$$

Once the cube map for each optical center is obtained, we go through the image using polar coordinates. For a fixed radius, we change θ and compute the pixel to color, obtained from Equation (Equation 15). Knowing θ and ϕ, from Table 5, we can cast a ray that goes from the catadioptric system to the cube map acquired. The intersection gives the information for the pixel.

In these non-central systems, the number of acquisitions required depends on the resolution of the final non-central image. That means, the more resolution the image has, the more acquisitions are needed. For an efficient composition of images, we need to define as fast as possible the pose of the camera in the virtual environment for each optical center. That is one of the reasons we use Unreal Engine 4 as a virtual environment, where we can easily change the camera pose in the virtual environment making fast acquisitions, since the graphics work in real time.

## 3. Results

Working on an environment of Unreal Engine 4 [12] and the simulator presented in this paper, we have obtained a variety of photorealistic omnidirectional images from different systems. In the Appendix A we have several examples of these images. To evaluate if our synthetic images can be used in computer vision algorithms, we compare the evaluation of four algorithms with our synthetic images and real ones. The algorithms selected are:**Corners For Layouts**: CFL [37] is a neural network that recovers the 3D layout of a room from an equirectangular panorama. We have used a pre-trained network to evaluate our images.**Uncalibtoolbox**: the algorithm presented in [34] is a MatLab toolbox for line extraction and camera calibration for different fish-eye and central catadioptric systems. We compare the calibration results from different images.**OpenVSLAM**: a virtual Simultaneous Location and Mapping framework, presented in [38], which allows use of omnidirectional central image sequences.**3D line Reconstruction from single non-central image** which was presented in [39,40] using non-central panoramas.

### 3.1. 3D Layout Recovery, CFL

Corners For layouts (CFL) is a neural network that recovers the layout of a room from one equirectangular panorama. This neural network provides two outputs: one is the intersection of walls or edges in the room and the second is the corners of the room. With those representations, we can build a 3D reconstruction of the layout of the room using Manhattan world assumptions.

For our evaluation, we have used a pre-trained CFL (https://github.com/cfernandezlab/CFL) network with Equirectangular Convolutions (*Equiconv*). The training dataset was composed by equirectangular panoramas built from real images and the ground truth was made manually, which increases the probability of mistakes due to human error. The rooms that compose this dataset are 4-wall rooms (96%) and 6-wall rooms (6%).

To compare the performance of CFL, the test images are divided into real images and synthetic images. In the set of real images, we have used the test images from the datasets STD2D3D [4], composed of 113 equirectangular panoramas of 4-wall rooms, and the SUN360 [3], composed of 69 equirectangular panoramas of 4-wall rooms and 3 of 6-wall rooms. The set of synthetic images are divided in panoramas from 4-wall rooms and from 6-walls rooms. Both sets are composed by 10 images taken on 5 different places in the environment in 2 orientations. Moreover, for the synthetic sets, the ground-truth information of the layout has been obtained automatically with pixel precision.

The goal of these experiments is testing the performance of the neural network in the different situations and evaluate the results using our synthetic images and comparing with those obtained with the real ones. In the figures above the ground-truth generation can be seen, Figure 7a,c and Figure 8a,c, and the output of CFL, Figure 7b,d and Figure 8b,d, for a equirectangular panorama in the virtual environments recreated. On the 4-wall layout environment we can observe that the output of CFL is similar to the ground truth. This seems logical since most of the images from the training dataset have the same layout. On the other hand, the 6-wall layout environment presents poorer results. The output from CFL in this environment only fits four walls of the layout, probably due to the gaps in the training data.

To quantitatively compare the results of CFL, in Table 6 we present the results using real images from existing datasets with the results using our synthetic images. We compare five standard metrics: Intersection over union of predicted corner/edge pixels (IoU), accuracy Acc, precision P, Recall R and F1 Score $F_{1}$.

### 3.2. Uncalibtoolbox

The uncalibtoolbox is a MatLab toolbox where we can compute a line extraction and calibration on fish-eye lenses and catadioptric systems. This toolbox makes the line extraction from the image and computes the calibration parameters from the distortion on these lines. The more distortion the lines present in the image, the more accurate the calibration parameters are computed. Presented in [34], this toolbox considers the projection models to obtain the main calibration parameter ${\hat{r}}_{vl}$ of the projection system. This ${\hat{r}}_{vl}$ parameter encapsulates the distortion of each projection and is related with the field of view of the camera.

On this evaluation we want to know if our synthetic images can be processed as real images on computer vision algorithms. We take several dioptric and catadioptric images generated by our simulator and perform the line extraction on them. To compare the results of the line extraction, we compare with real images from [34].

In the previous Figures, we presented four equi-angular fish-eye, Figure 9, and two catadioptric systems, Figure 10. The behavior of the line extraction algorithm of uncalibtoolbox for the synthetic images of our simulator is similar to the real images presented in [34]. That encourages our work, because we have made photorealistic images that can be used as real ones for computer vision algorithms.

On the other hand, we compare the accuracy of the calibration process between the results presented in [34] and the obtained with our synthetic images. The calibration parameter has been obtained testing 5 images for each $r_{vl}$ and taking the mean value. Since we impose the calibration parameters in our simulator, we have selected 10 values of the parameter $r_{vl}$, in the range $0.5<r_{vl}<1.5$, in order to compare our images with the results of [34]. The calibration results are presented in the Figure 11.

### 3.3. OpenVSLAM

The algorithm presented in [38] allows the obtaining of a SLAM for different cameras, from perspective to omnidirectional central systems. This open-source algorithm is based on an indirect SLAM algorithm, such as ORB-SLAM [41] and ProSLAM [42]. The main difference with other SLAM approaches is that the proposed framework allows definition of various types of central cameras other than perspective, such as fish-eye or equirectangular cameras.

To evaluate the synthetic images generated with OmniSCV, we create a sequence in a virtual environment simulating the flight of a drone. Once the trajectory is defined, we generate the images where we have the ground truth of the pose of the drone camera.

The evaluation has been made with equirectangular panoramas of 1920×960 pixels through a sequence of 28 seconds and 30 frames per second. The ground-truth trajectory as well as the scaled SLAM trajectory can be seen in Figure 12. In the Appendix B we include several captures from the SLAM results as well as the corresponding frame obtained with OmniSCV. We evaluate quantitatively the precision of the SLAM algorithm computing the position and orientation error of each frame respect to the ground-truth trajectory. We consider error in rotation, $\varepsilon_{\theta }$, and in translation, $\varepsilon_{t}$, as follows:(16)$$\varepsilon_{t}=\arccos\left(\frac{t_{gt}^{T}\cdot t_{est}}{|t_{gt}\left|\right|t_{est}|}\right);\;\;\varepsilon_{\theta }=\arccos\left(\frac{Tr\left(R_{gt}{R_{est}}^{T}\right)-1}{2}\right),$$
where $t_{gt},R_{gt}$ are the position and rotation matrix of a frame in the ground-truth trajectory and $t_{est},R_{est}$ are the estimated position up to scale and rotation matrix of the SLAM algorithm in the same frame. The results of these errors are shown in Figure 13.

### 3.4. 3D Line Reconstruction from Single Non-Central Image

One of the particularities of non-central images is that line projections contain more geometric information. In particular, the entire 3D information of the line is mapped on the line projection [43,44].

For evaluating if synthetic non-central images generated by our tool conserve this property, we have tested the proposal presented in [40]. This approach assumes that the direction of the gravity is known (this information could be recovered from an inertial measurement unit (IMU)) and lines are arranged in vertical and horizontal lines. Horizontal lines can follow any direction contained in any horizontal plane (soft-Manhattan constraint). The non-central camera captures a non-central panorama of 2048 × 1024 pixels with a radius of $R_{c}=1m$ and an inclination of 10 degrees from the vertical direction.

A non-central-depth synthetic image has been used as ground truth of the reconstructed points (see Figure 14b). In Figure 14a we show the extracted line projections and segments on the non-central panoramic image; meanwhile, Figure 15 presents the reconstructed 3D line segments. We have omitted the segments with low effective baseline in the 3D representation for visualization purposes.

## 4. Discussion

From our tool we have obtained RGB, depth and semantic images from a great amount of omnidirectional projection systems. These images have been obtained from a photorealistic virtual world where we can define every parameter. To validate the images obtained from our tool, we have made evaluations with computer vision algorithms that use real images.

In the evaluation with CFL we have interesting results. On one hand, we have obtained results comparable to datasets with real images. This behavior shows that the synthetic images generated with our tool are as good as real images from the existing datasets. On the other hand, we have made different tests changing the layout of our scene, something that cannot be done in real scenarios. On these changes we have realized that CFL does not work properly with some layouts. This happens because existing datasets have mainly 4-wall rooms to use as training data and the panoramas have been taken in the middle of the room [2,4]. This makes it hard for the neural network to generalize for rooms with more than 4 walls or panoramas that have been taken in different places inside the room. Our tool can aid in solving this training problem. Since we can obtain images from every place in the room and we can change the layout, we can fill the gaps of the training dataset. With bigger and richer datasets for training, neural networks can improve their performance and make better generalizations.

In the evaluation with uncalibtoolbox, we have tested catadioptric systems and fish-eye lenses. We have compared the precision of the toolbox for real and synthetic images. In the line extraction, the toolbox has no problems nor makes any distinction from one kind of images or the other. That encourages our assumptions that our synthetic images are photorealistic enough to be used as real images. When we compare the calibration results, we can see that the results of the images obtained from [34] and the results from our synthetic images are comparable. There are no big differences in precision. The only different values observed are in hyper-catadioptric systems. For the hyper-catadioptric systems presented in [34], the computed calibration parameters differ from the real ones while in the synthetic hyper-catadioptric systems, we have more accurate parameters. A possible conclusion of this effect is the absence of the reflection of the camera in the synthetic images. For those, we have more information of the environment in the synthetic images than in real ones, helping the toolbox to obtain better results for the calibration parameters. From the results shown, we can conclude that our tool can help to develop and test future calibration tools. Since we are the ones that set the calibration of the system in the tool, we have perfect knowledge of the calibration parameters of the image. However, real systems need to be calibrated a priori or we must trust the calibration parameters that the supplier of the system gives us.

In the evaluation of the SLAM algorithm, we test if the synthetic images generated with our tool can be used in computer vision algorithms for tracking and mapping. If we compare the results obtained from the OpenVSLAM algorithm [38], with the ground-truth information that provides our tool, we can conclude that the synthetic images generated with OmniSCV can be used for SLAM applications. The position error is computed in degrees due to the lack of scale in the SLAM algorithm. Moreover, we observe the little position and orientation error of the camera along the sequence (see Figure 13), keeping the estimated trajectory close to the real one. Both errors are less than 8 degrees and decrease along the trajectory. This correction of the position is the effect of the loop closure of the SLAM algorithm. On the other hand, we obtain ground-truth information of the camera pose for every frame. This behavior encourages the assumptions we have been referring to in this section: that synthetic images generated from our tool can be used as real ones in computer vision algorithms, obtaining more accurate ground-truth information too.

Finally, in the evaluation of the non-central 3D line fitting from single view we can see how the non-central images generated with our tool conserve the full projection of the 3D lines of the scene. It is possible to recover the metric 3D reconstruction of the points composing these lines. As presented in [39] this is only possible when the set of projecting skew rays composing the projection surface of the segment have enough effective baseline.

## 5. Conclusions

In this work, we present a tool to create omnidirectional synthetic photorealistic images to be used in computer vision algorithms. We devise a tool to create a great variety of omnidirectional images, outnumbering the state of the art. We include in our tool different panoramas such as equirectangular, cylindrical and non-central; dioptric models based on fish-eye lenses (equi-angular, stereographic, orthogonal and equi-solid angle); catadioptric systems with different kinds of mirrors as spherical, conical, parabolic and hyperbolic; and two empiric models, Scaramuzza’ and Kannala–Brandt’s. Moreover, we get not only the photorealistic images but also labeled information. We obtain semantic and depth information for each of the omnidirectional systems proposed with pixel precision and can build specific functions to obtain ground truth for computer vision algorithms. Furthermore, the evaluations of our images show that we can use synthetic and real images equally. The synthetic images created by our tool are good enough to be used as real images in computer vision algorithms and deep-learning-based algorithms.

## Figures and Tables

**Figure 1 sensors-20-02066-f001:**
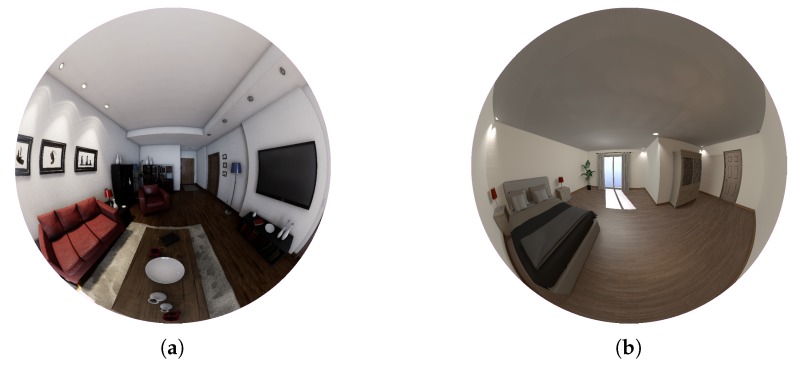
(**a**): Kannala–Brandt projection obtained from Unreal Engine 4. (**b**): Kannala–Brandt projection obtained from POV-Ray.

**Figure 2 sensors-20-02066-f002:**
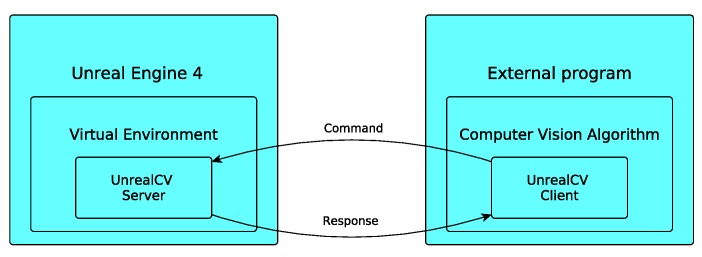
Client–server communication between Unreal Engine 4 and an external program via UnrealCV.

**Figure 3 sensors-20-02066-f003:**
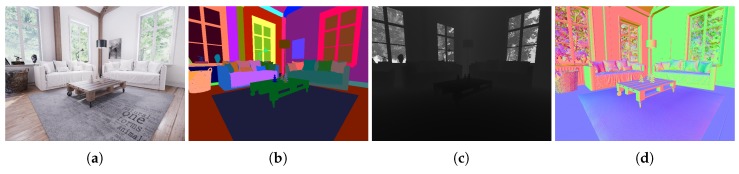
Capture options available in UnrealCV. (**a**): Lit RGB image; (**b**): Object mask; (**c**): Depth; (**d**): Surface normal.

**Figure 4 sensors-20-02066-f004:**
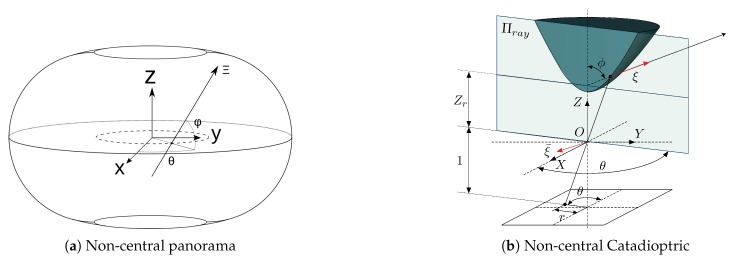
Non-central systems schemes.

**Figure 5 sensors-20-02066-f005:**
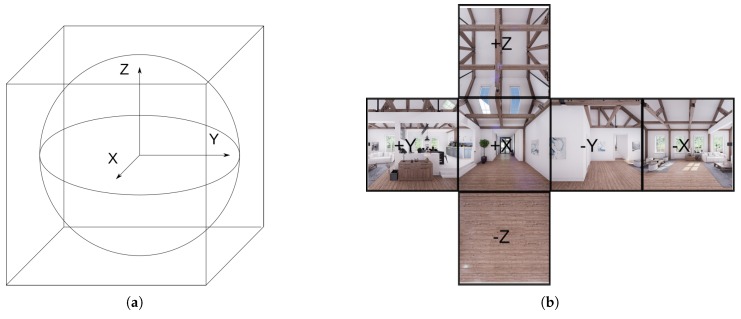
(**a**): Simplification of the sphere into the cube map; (**b**): Unfolded cube map from a scene.

**Figure 6 sensors-20-02066-f006:**
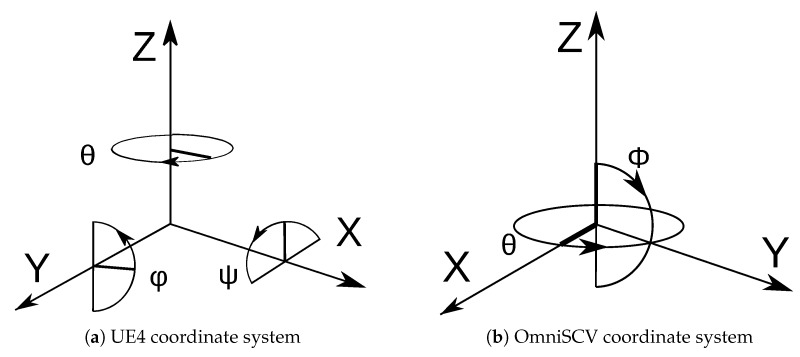
(**a**): Coordinate system used in graphic engines focused on first-person video games; (**b**): Coordinate system of our image simulator.

**Figure 7 sensors-20-02066-f007:**
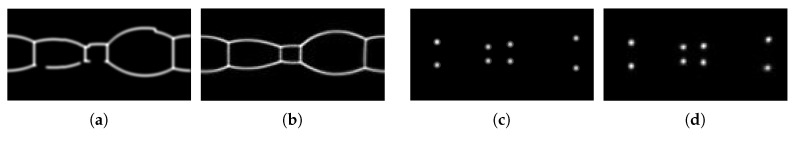
Results of CFL using a synthetic image from a 4-wall environment. (**a**): Edges ground truth; (**b**): Edges output from CFL; (**c**): Corners ground truth; (**d**): Corners output from CFL.

**Figure 8 sensors-20-02066-f008:**
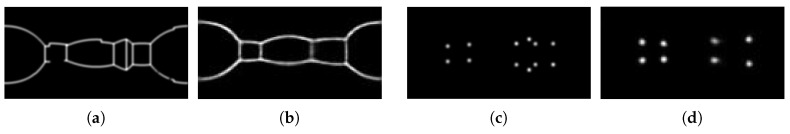
Results of CFL using a synthetic image from a 6-wall environment. (**a**): Edges ground truth; (**b**): Edges output from CFL; (**c**): Corners ground truth; (**d**): Corners output from CFL.

**Figure 9 sensors-20-02066-f009:**
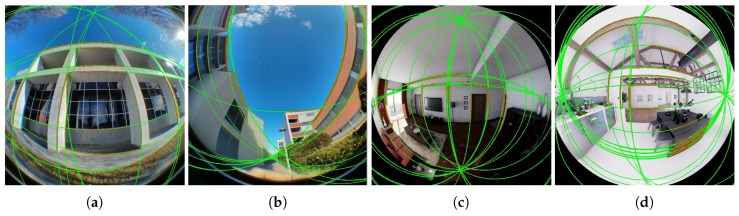
Line extraction on fish-eye camera. (**a**,**b**): Real fisheye images from examples of [34]; (**c**,**d**): Synthetic images from our simulator.

**Figure 10 sensors-20-02066-f010:**
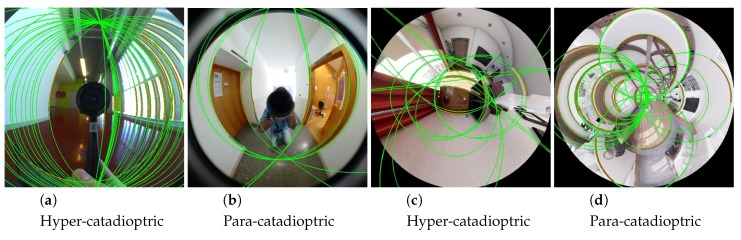
Line extraction on catadioptric systems. (**a**,**b**): Real catadioptric images from examples of [34]; (**c**,**d**): Synthetic images from our simulator.

**Figure 11 sensors-20-02066-f011:**
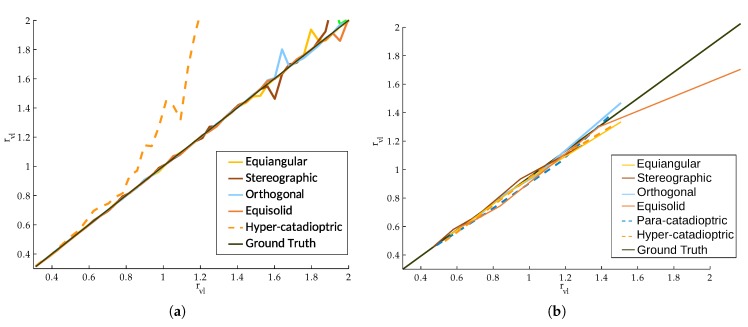
Normalized result for the calibration parameters using different omnidirectional cameras. (**a**): Calibration results from [34]; (**b**): Calibration results using images from our simulator.

**Figure 12 sensors-20-02066-f012:**
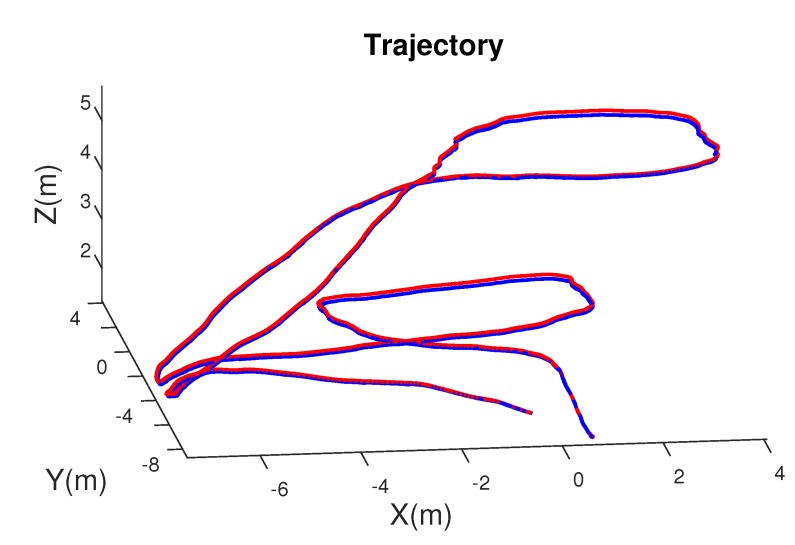
Visual odometry from SLAM algorithm. The red line is the ground-truth trajectory while the blue line is the scaled trajectory of the SLAM algorithm.

**Figure 13 sensors-20-02066-f013:**
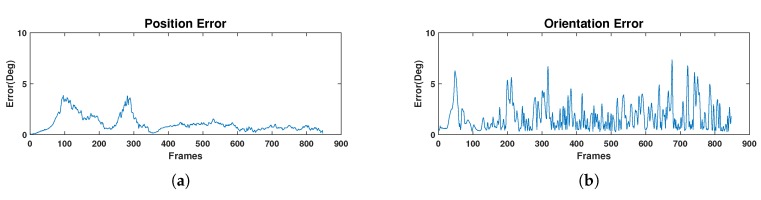
(**a**): Position error of the SLAM reconstruction. (**b**): Orientation error of the SLAM reconstruction. Both errors are measured in degrees.

**Figure 14 sensors-20-02066-f014:**
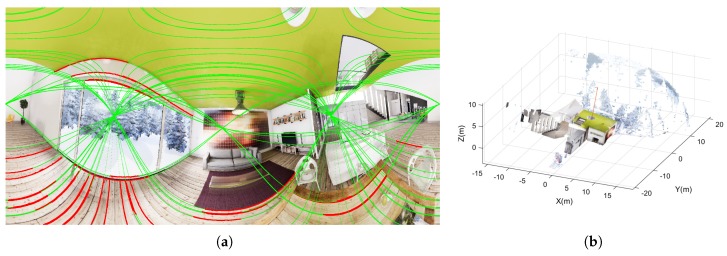
(**a**) Extracted line projections and segments on the non-central panorama. (**b**) Ground-truth point-cloud obtained from depth-map.

**Figure 15 sensors-20-02066-f015:**
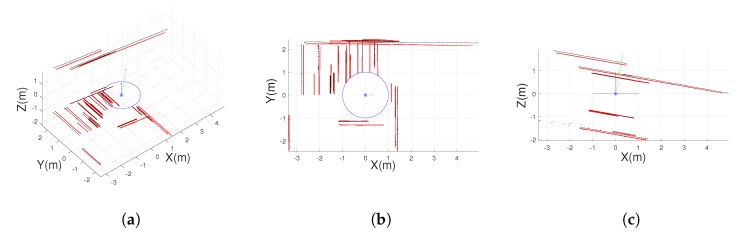
3D line segments reconstructed from line extraction in non-central panorama. In red the reconstructed 3D line segments. In black the ground truth. In blue the circular location of the optical center and the Z axis. In green the axis of the vertical direction. (**a**) Orthographic view. (**b**) Top view. (**c**) Lateral view.

**Table 1 sensors-20-02066-t001:** Definition of $h\left(\phi \right)=\hat{r}$ for different fish-eye lenses.

Equi-Angular	Stereographic	Orthogonal	Equi-Solid Angle
fϕ	2ftan(ϕ/2)	fsinϕ	fsin(ϕ/2)

**Table 2 sensors-20-02066-t002:** Definition of Ψ and ξ for different mirrors.

Catadioptric System	ξ	Ψ
Hyper-catadioptric	0<ξ<1	$\frac{d+2p}{\sqrt{d^{2}+4p^{2}}}$
Para-catadioptric	1	1+2p

**Table 3 sensors-20-02066-t003:** Relationship between $\hat{r}$ and ϕ from the fish-eye projection model.

Equi-Angular	Stereographic	Orthogonal	Equi-Solid Angle
$\phi =\hat{r}/f$	$\phi =2\arctan(\hat{r}/2f)$	$\phi =\arcsin(\hat{r}/f)$	$\phi =2\arcsin(\hat{r}/f)$

**Table 4 sensors-20-02066-t004:** Definition of ξ and η for central mirrors.

Catadioptric System	ξ	η
Para-catadioptric	1	−2p
Hyper-catadioptric	$\frac{d}{\sqrt{d^{2}+4p^{2}}}$	$\frac{-2p}{\sqrt{d^{2}+4p^{2}}}$

**Table 5 sensors-20-02066-t005:** Definition of cotϕ and $Z_{r}$ for different mirrors.

Mirror	$Z_{r}$	cotϕ
Conical	$Z_{c}+R_{c}\cot\phi$	(1+rtan(2τ))/(tan(2τ)−r)
Spherical	−ξ/δ	$Z_{s}\left(\delta +\epsilon \right)/\delta$

**Table 6 sensors-20-02066-t006:** Comparative of results of images from different datasets. *OmniSCV* contains the images created with our tool on a 6-wall room and a 4-wall room. The real images have been obtained from the test dataset of [2,4].

Test Images	Edges						Corners				
	IoU	Acc	P	R	$F_{1}$		IoU	Acc	P	R	$F_{1}$
6-wall OmniSCV	0.424	0.881	0.694	0.519	0.592		0.370	0.974	0.547	0.526	0.534
4-wall OmniSCV	0.474	0.901	0.766	0.554	0.642		0.506	0.985	0.643	0.698	0.669
Real images	0.452	0.894	0.633	0.588	0.607		0.382	0.967	0.758	0.425	0.544

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
