# Peer review of "OmniSCV: An Omnidirectional Synthetic Image Generator for Computer Vision"

_sensors, 2020, doi:10.3390/s20072066_

Round 1

Reviewer 1 Report

The authors present a framework for synthetic image generation. The images are generated from Unreal Tournament game engine, through a third-party plugin, named UnrealCV.

The description of the work is well detailed and has some limited discussion also. The main contribution is the integration work, which is a nice engineering work, with little scientific part.

The main part, which is lacking from the paper is the description of the novelty. The projection models presented in 2.2 are not new and have little scientific contribution. It would beneficial if the authors would describe, the added value to the existing background.

For example: the work presented in 2.3.1 is a work initiated by Paul Bourke, originated back to 2003 (http://paulbourke.net/miscellaneous/cubemaps/).

What did the authors achieve than implementing this?

Is the OmniSCV available somewhere to be downloaded?

Any ideas around processing costs? Experimental setup? Comparison of real vs. virtual scenes?

Reviewer 2 Report

This work presents the construction of synthetic omni images with different sort of projections. Authors should emphasize and clarify that this procedure acquires images from a particular framework (by specifically naming it). A more straightforward and brief synthesis of details about the implementation should be also included in the abstract for disambiguation of such aspect. The validation scope of these generated images is too narrow. Authors should concentrate on its extension to open-ended and widely accepted approaches.

Specifically, I suggest authors to cover these comments:

It should be clearly addressed in the abstract what this work has performed: an utilization of the images provided by Unreal Engine in order to construct catadioptric scenes. Please take this into account. The abstract only refers to "a common framework". Written like now, it might seem that pretends this is an entire and custom implementation, but it's not. Please reword. Also, bold highlighting doesn't look standard for an abstract.

Why Unreal and not other dataset? Please discuss if this tool would be easily adapted to other datasets and made it available.

My main concern falls on the results section. Authors present a test with two own computer vision oriented approaches ([31,23]. Nonetheless, this validation should be extended to some other, widely acknowledged, and open target algorithms. Final end applications should be tested and compared by using this sort of images. It's crucial that other publicly available methods are explored (ie: for image localization, visual odo, or either visual SLAM). An example to compare with would be some kind of synthetic dataset like these:

http://rpg.ifi.uzh.ch/fov.html

Another point of concern is that almost all the projection models are referred to own works [27,28,29], either because they were implemented in such works (fact that fades the reader's perception of contribution of the paper), or either because it's simply a secondary reference. If so, at least more relevant cites for such theory models of projection should be included instead than own cites.

Some english statements and expresion could be more explicitly clear, aiming at better readability.

l1: "the aim is the need...". The scope and goal of a scientific work shouln't be a need. Please concentrate on the sort of implementation conducted

l7: "the omnidirectional images". This should be a general term so try to avoid the use of article. Same comment along the entire text.

l22. makes

Scaramuzza, Kannala, Unreal Engine, etc...should be cited from the very first time they are discussed in the Intro.

l78. "or make". If noun should be listed as a noun, otherwise is it a verb? to make.

l299. recovers

Round 2

Reviewer 1 Report

The authors have improved the quality of their work and have successfully answered the questions raised by the reviewers.

Reviewer 2 Report

I congratulate authors for their work on achieving this integration implementation and for their effort dedicated to address my comments.

I assume that reviewing and modifying the paper in a rush, made authors forget to check that latex compiler failed to include references, cites to them, and also references to figures/tables. Please take time to confirm that are correctly generated and included in the PDF version.